# Sequential Hypofractionated versus Concurrent Twice-Daily Radiotherapy for Limited-Stage Small-Cell Lung Cancer: A Propensity Score-Matched Analysis

**DOI:** 10.3390/cancers14163920

**Published:** 2022-08-13

**Authors:** Wei Zhou, Pang Wang, Xinyu Ti, Yutian Yin, Shigao Huang, Zhi Yang, Jie Li, Guangjin Chai, Bo Lyu, Zhaohui Li, Yan Zhou, Feng Xiao, Lin Xu, Mei Shi, Lina Zhao

**Affiliations:** 1Department of Radiation Oncology, Xijing Hospital, Air Force Medical University, Xi’an 710032, China; 2Out-Patient Department, Xijing Hospital, Air Force Medical University, Xi’an 710032, China; 3Department of Pulmonary Diseases and Critical Care Medicine, Xijing Hospital, Air Force Medical University, Xi’an 710032, China

**Keywords:** LS-SCLC, sequential hypofractionated, radiotherapy, propensity score, concurrent twice-daily

## Abstract

**Simple Summary:**

Although twice-daily concurrent chemoradiotherapy for limited-stage small-cell lung cancer (LS-SCLC) is still the standard treatment, this regimen is inconvenient and not universally adopted across different institutions. Therefore, the optimal radiation dose and fractions are still under investigation. A once-daily hypofractionated schedule may be suitable for LS-SCLC patients, and more and more hypofractionated schedule studies are being performed. However, few studies have investigated the efficacy and toxicities of sequential chemotherapy and hypofractionated radiotherapy compared with the concurrent twice-daily schedule. Our evaluation of sequential hypofractionated and twice-daily concurrent chemoradiotherapy schedules before and after propensity score-matched analysis (PSM) for LS-SCLC revealed a comparable survival and less toxicity. The sequential hypofractionated schedule may be used as an alternative to concurrent twice-daily regimens.

**Abstract:**

Background: As there are no randomized trials comparing twice-daily with sequential hypofractionated (sequential hypo) radiotherapy regimens for limited-stage small-cell lung cancer (LS-SCLC). This study aimed to compare these two regimens for LS-SCLC by propensity score-matched analysis (PSM). Methods: We retrospectively analyzed 108 LS-SCLC patients between January 2015 and July 2019. All patients received concurrent twice-daily or sequential hypo radiotherapy. The survival, failure patterns, and toxicities were evaluated before and after PSM. Results: Before PSM, multivariate analysis showed that patients treated with sequential hypo had a significantly better overall survival (OS) and distant metastasis-free survival (DMFS) (HR = 0.353, *p* = 0.009; HR = 0.483, *p* = 0.039, respectively). Total radiotherapy time ≥ 24 days and stage III (HR = 2.454, *p* = 0.004; HR = 2.310, *p* = 0.004, respectively) were poor prognostic indicators for OS. Patients with a total radiotherapy time ≥ 24 days and N2–3 were more likely to recur than others (HR = 1.774, *p* = 0.048; HR = 2.369, *p* = 0.047, respectively). N2–3 (HR = 3.032, *p* = 0.011) was a poor prognostic indicator for DMFS. After PSM, being aged ≥65 years was associated with poorer OS, relapse-free survival (RFS) and DMFS (*p* < 0.05). A total radiotherapy time of ≥24 days was a poor prognostic indicator for OS and RFS (HR = 2.671, *p* = 0.046; HR = 2.370, *p* = 0.054, respectively). Although there was no significant difference, the patients in the sequential hypo group had a trend towards a better OS. The failure pattern between the two groups showed no difference. More patients had grade 1–2 esophagitis in the twice-daily group (*p* = 0.001). Conclusions: After propensity matching, no difference was shown in survival and failure. The sequential hypo schedule was associated with comparable survival and less toxicity and may be used as an alternative to concurrent twice-daily regimens.

## 1. Introduction

Lung cancer remains a malignant tumor with high morbidity and mortality and it is the leading cause of cancer-related death worldwide [1]. Small-cell lung cancer (SCLC) has been calculated to represent about 15% of all lung cancers, which is characterized by its highly invasive capacity and early metastatic behavior [2]. About one-third of SCLC patients are diagnosed with limited-stage small-cell lung cancer (LS-SCLC) and they have the potential for a cure [3]. Although highly sensitive to chemoradiotherapy, the prognosis of LS-SCLC is still very poor, with a median survival time of only 15 to 20 months [4,5].

ECOG INT0096 [6] indicated that the survival for patients receiving twice-daily radiotherapy (45 Gy/30 F) with concurrent cisplatin/etoposide (EP) chemotherapy was significantly improved compared to the once-daily regimen (45 Gy/25 F). Currently, the EP regimen combined with early intervention of concurrent twice-daily thoracic radiation therapy (TRT) for LS-SCLC remains the standard treatment [7,8,9]. The optimal radiation dose and fractions are still under investigation. The CONVERT study showed that the survival for LS-SCLC patients did not differ between twice-daily (45 Gy, 1.5 Gy per fraction) and once-daily (66 Gy, 2 Gy per fraction) concurrent chemoradiotherapy [10]. Considering the twice-daily schedule is not convenient for patients and is not universally adopted across different institutions, the once-daily hypofractionated schedule may be suitable for LS-SCLC patients. A retrospective analysis showed that hypofractionated thoracic radiotherapy (HypoTRT) (55 Gy, 2.5 Gy per fraction daily) or conventional thoracic radiotherapy (ConvTRT) (median dose 60 Gy, 2 Gy per fraction daily) combined with the EP regimen had similar survival outcomes and toxicities [11]. In a randomized phase II trial of 157 cases of LS-SCLC comparing concurrent HypoTRT (42 Gy/15 F, once daily) and a twice-daily (45 Gy/30 F in 30 fractions, BID) schedule, there was no significant difference in survival outcomes and toxicities [12]. A recent phase II randomized study indicated that concurrent hypofractionated radiotherapy (65 Gy in 26 fractions) improved PFS and had similar toxicities compared with a concurrent twice-daily regimen in LS-SCLC [13]. In another overlap-weighted analysis, there was no significant difference in overall survival, local control, and toxicity for LS-SCLC patients treated with hypofractionated radiotherapy versus the twice-daily group [14]. However, few studies have investigated the efficacy and toxicities of sequential chemotherapy and hypofractionated radiotherapy compared to the concurrent twice-daily schedule.

In this study, we evaluated survival, toxicities, and failure patterns between a concurrent twice-daily schedule with sequential hypofractionated radiotherapy (sequential hypo) in LS-SCLC by PSM analysis. Furthermore, we believe that the sequential hypo could be an alternative option for LS-SCLC treatment without increasing toxicity.

## 2. Materials and Methods

### 2.1. Patients

We retrospectively collected the data of 108 patients eligible for enrollment (79 in the concurrent twice-daily group and 29 in the sequential hypo radiotherapy group); all patients were from the Xijing Hospital between January 2015 and July 2019. They were histologically or cytologically diagnosed with SCLC and the initial staging determined LS-SCLC. Routine examinations included hematological examination, contrast-enhanced chest and abdominal CT scans, lymph node ultrasound, bone scan, and an MRI scan of the brain were tested before treatment. Whole-body PET-CT was performed at the patient’s discretion. The initial staging of LS-SCLC was according to the Veterans Administration Lung Study Group (VALSG) two-stage classification [15] and the American Joint Committee on Cancer (AJCC) stage [16]. Patients who had undergone surgery and were not able to tolerate radiation therapy were excluded from this study. Informed consent was obtained from all patients.

### 2.2. Radiotherapy

We used the Varian IX linear accelerator with an energy of 6 MV X-ray and a dose rate of the radiation adjustable from 0 to 600 MU/min in this study. Radiation therapy was delivered with volumetric intensity modulated arc therapy (VMAT). In the delivery of radiation, patients were immobilized in a body membrane fixation in the supine position, and then scanned with a CT slice thickness of 5 mm. The gross tumor volume (GTV) was contoured restricted to the visible primary tumor and positive lymph nodes based on CT. The clinical target volume (CTV) was delineated to cover 6 mm surrounding the GTV and involved node regions. An expansion of 5 mm of the CTV created a planning target volume (PTV). Patients received either concurrent twice-daily radiotherapy (45 Gy/30 F, twice daily, with a minimum of six hours between fractions) or sequential hypo radiotherapy (total dose ≥ 45 Gy, 3 Gy per fraction, once daily) according to patient and physician choice. The prescribed doses of sequential hypo radiotherapy were 45 Gy in 15 fractions to CTV, and 54 Gy in 18 fractions or 60 Gy in 20 fractions for GTV and GTVnd. It could be administered two weeks after the end of the last chemotherapy cycle. In addition, the timing of thoracic radiotherapy in the CCRT setting was on the third or fourth chemotherapy cycle. The treatment was performed on consecutive weekdays. Chemotherapy with etoposide and cisplatin (etoposide 100 mg/m^2^ intravenously on days 1–3 and cisplatin 75 mg/m^2^ intravenously on day 1) was administered every three weeks with up to four to six cycles, and the dosage could be adjusted according to toxicities. Prophylactic cranial irradiation (PCI) could be started 3–4 weeks after the completion of thoracic radiotherapy in patients with a complete response or partial response. PCI was performed with three-dimensional conformal radiotherapy (3D-CRT), and the dose was 25 Gy in 10 fractions. All treatments were approved by the ethics committee prior to its clinical application.

### 2.3. Acute and Late Toxicities

Radiotherapy-related toxicity was evaluated according to the Acute and Late Radiation Morbidity Scoring Criteria of RTOG. Chemotherapy-related toxicities were evaluated by the Common Terminology Criteria for Adverse Events (CTCAE) version 3.0 of the National Cancer Institute (NCI). The patients’ acute and late toxicities were evaluated after radiotherapy. The body examination included a comprehensive hematological examination, ultrasound examination, and contrast-enhanced chest CT scanning. A brain magnetic resonance imaging (MRI) scan could be performed if the patient had suspicious cranial symptoms. The follow-up times were generally every 3 months for the first year and every 3–6 months for the next 2 years.

### 2.4. Propensity Score Matching

PSM was used to balance the baseline confounders of patient characteristics between the concurrent twice-daily and sequential hypo radiotherapy groups by adjusting the variables of age, T stage, N stage, PCI, and Eastern Cooperative Oncology Group performance status (ECOG PS). The propensity score was defined as the probability of receiving treatment (concurrent twice-daily or sequential hypo radiotherapy) based on the conditions of adjusted factors. Propensity scores [17,18] were generated using multivariable logistic regression models, in which the group was used as the outcome variable and the adjusted factors as predictors. The most common straightforward technique of 1:1 nearest neighbor matching was used in this matching. To ensure good matches, we selected a caliper (maximum allowable difference between two participants) of 0.2.

### 2.5. Statistical Analysis

The follow-up period was measured from the beginning of treatment to the time of death or the last follow-up before analysis. The overall survival (OS) was defined as the time from diagnosis to death from any cause or the last follow-up. The relapse-free survival (RFS) was defined as the time from diagnosis until relapse (local or regional disease recurrence) or death from any cause. The distant metastasis-free survival (DMFS) was defined as the time from diagnosis to distant metastasis or death from any cause. The continuous variables, such as age, the date of the start of treatment to the end of radiotherapy (SER), total chemotherapy cycles, chemotherapy cycles before radiotherapy, and total radiotherapy time, did not conform to a normal distribution. We used M (QL, QU) to indicate its centralized and discrete trends. The differences between the two groups were detected by Mann–Whitney U. Receiver operating characteristic (ROC) curves were used to convert the continuous variables into two subgroups at their cutoff values identified by overall survival (OS). The Wilcoxon rank sum test or chi-square test were used to compare the categorical variables (expressed in frequency or percentage). The Kaplan–Meier curve was used to calculate actuarial rates of the percentage survival, OS, RFS, as well as DMFS. The differences in the time-to-event outcomes between groups were compared by logrank tests.

Univariable and multivariable Cox analyses were used to detect the influence of associated clinical factors for OS, RFS, and DMFS. The nomograms were established to predict the combined effect of the influencing factors on OS, RFS, and DMFS based on the multivariate Cox regression analysis. A two-tailed *p* value < 0.05 was considered statistically significant. The SPSS (version 24.0, IBM, Armonk, NY, USA) and R 3.6.1 software (www.R-project.org, The R Project for Statistical Computing, Vienna, Austria) were used for statistical analyses.

## 3. Results

### 3.1. Patient Characteristics and Treatment

The baseline of patient characteristics in the two groups (“before PSM (*n* = 108)” and “after PSM (*n* = 48)”) were shown in Table 1. The SER, chemotherapy cycles before radiotherapy, and total radiotherapy time in the hypofractionated group were significantly longer than for patients in the concurrent twice-daily group in the data set of “before PSM” due to the choice of different modalities. PSM was used by adjusting the variables of age, T stage, N stage, PCI, and ECOG PS. After PSM, the confounders of patient characteristics between the concurrent twice-daily and hypofractionated group in the best final match of 24 pairs were well balanced (age, *p* = 0.718; T staging, *p* = 0.149; N staging, *p* = 0.712; PCI, *p* = 0.755; ECOG PS, *p* = 1.000).

### 3.2. Survival and Prognostic Factors

The median follow-up time was 23.9 months (range, 0.83–66.5 months). Fifty-seven patients were alive and 51 died. The median follow-up time of survivors was 34.6 months (range: 10.5–66.5 months). There was no significant difference in OS, RFS, and DMFS between the concurrent twice-daily group and the sequential hypo group (*p* = 0.076, *p* = 0.107, *p* = 0.079, respectively) (Figure 1a,c,e). However, there was a trend that the sequential hypo group had a longer OS and DMFS than the concurrent twice-daily group.

A univariable cox analysis of OS, RFS, and DMFS before PSM is shown in Appendix A. Patients treated with a total radiotherapy time of ≥24 days had a significantly shorter OS and RFS than those treated with a total radiotherapy time of <24 days (HR =1.934, 95%CI = 1.072–3.488, *p* = 0.028; HR = 1.857, 95%CI = 1.051–3.280, *p* = 0.033, respectively). Patients who were former or current smokers were more likely to suffer a recurrence than those who were never smokers (HR = 2.368, 95% CI = 1.000–5.612, *p* = 0.050; HR = 2.366, 95%CI = 1.029–5.437, *p* = 0.043, respectively). N of stage 2–3 (HR = 2.742, *p* = 0.020; HR = 2.472, *p* = 0.037, respectively) and AJCC III (HR = 2.696, *p* = 0.022; HR = 2.374, *p* = 0.046, respectively) were poor prognostic indicators for DMFS and RFS.

In multivariate analysis before PSM (Appendix A), being in the sequential hypo group was a protective factor for OS and DMFS (HR = 0.353, *p* = 0.009; HR = 0.483, *p* = 0.039). A total radiotherapy time of ≥24 days and AJCC stage of III were poor prognostic indicators for OS (Appendix A) (HR = 2.452, *p* = 0.004; HR = 2.310, *p* = 0.055, respectively). Patients with a total radiotherapy time of ≥24 days were more likely to suffer recurrence than others (HR = 1.774, *p* = 0.048). An N stage of 2–3 was a poor prognostic indicator for RFS (Appendix A) and DMFS (Appendix A) (HR = 2.369, *p* = 0.047; HR = 3.032, *p* = 0.011, respectively).

### 3.3. Survival and Prognostic Factors for Matched Patients

After PSM, the sequential hypo group had a trend towards a better OS and DMFS than the concurrent twice-daily group, but there was no significant difference in OS between the two groups (HR = 0.632, *p* = 0.377; HR = 0.569, *p* = 0.231, respectively) (Figure 1b,f). At the same time, there was no significant difference in RFS between the two groups (HR = 0.697, *p* = 0.429) (Figure 1d).

In univariable Cox regression analyses (Table 2), we found that being aged ≥65 years, an SER of ≥110 days, and a total radiotherapy time of ≥24 days were associated with a significant decrease in OS (HR = 4.483, *p* = 0.002; HR = 3.124, *p* = 0.049; HR = 2.848, *p* = 0.030, respectively). Patients aged ≥65 years and with a total radiotherapy time of ≥24 days were more likely to suffer a recurrence than others (HR = 3.528, *p* = 0.007; HR = 2.478, *p* = 0.041, respectively). Being aged ≥65 years was a poor prognostic indicator in DMFS (HR = 3.383, *p* = 0.009).

In multivariable Cox regression analyses (Table 3), being aged ≥65 years was associated with a reduction in OS (Appendix A), RFS (Appendix A), and DMFS (HR = 4.222, *p* = 0.004; HR = 3.371, *p* = 0.010; HR = 3.383, *p* = 0.009, respectively). Having a total radiotherapy time of ≥24 days was a poor prognostic factor associated with OS (Supplemental Appendix A) and RFS (Appendix A) (HR = 2.671, *p* = 0.046, HR = 2.370, *p* = 0.054, respectively).

### 3.4. Failure Pattern

Before PSM, more distant metastasis occurred in the twice-daily group (40.5% vs. 17.7%, *p* < 0.05), and there was no significant difference in the locoregional failure rate. At the time of the last follow up, 57 patients were alive without any documented progression, and among the 51 deaths, 47 were related to SCLC, two died from a cause attributable to their comorbid conditions, none died from secondary cancer, and two died for unknown reasons. After PSM, there was no difference in failure rate between the two groups. Before and after PSM, the most common site of metastasis in the twice-daily group was the brain, followed by the liver and bone. The rate of distant metastasis appeared to be lower in the sequential hypo group, although there was no statistically significant difference (Table 4).

### 3.5. Toxicities

For all patients and matched patients in the sequential hypo group, no serious grade 4 adverse reactions were observed. Acute and late adverse events between the concurrent twice-daily and the sequential hypo group before and after PSM presented no significant difference (Table 5), except for hematotoxicity and esophagitis. Before PSM, more patients in the twice-daily group had ≥grade 3 hematotoxicity (25.3% vs. 3.4% in the sequential hypo group; *p* = 0.022). After PSM, the incidence of grade 1–2 esophagitis was 91.7% (22/24) in the twice-daily and 79.2% (19/29) in the once-daily group (*p* < 0.001). There were no deaths due to acute and late adverse events.

## 4. Discussion

Although twice-daily concurrent chemoradiotherapy is still the standard treatment for LS-SCLC [9,10], this regimen is inconvenient and is not universally adopted across different institutions. Therefore, the optimal radiation dose and fractions are still under investigation. Our evaluation of sequential hypo and twice-daily concurrent chemoradiotherapy regimens before and after PSM for LS-SCLC revealed a comparable survival and less toxicity, and this sequential hypo regimen may be considered for application across different institutions. To our knowledge, this is the first study to compare concurrent twice-daily with sequential hypofractionated radiotherapy (sequential hypo) for LS-SCLC by PSM analysis.

To date, there has been no consensus on the optimal radiation dose and fractions. A survey of U.S. radiation oncologists on practice patterns showed that three-quarters of the oncologists administered once-daily TRT (thoracic radiotherapy) more commonly than twice-daily regimen [19]. Because once-daily TRT is more convenient (71%), logistically easier for clinic schedules (43%), and often more tolerable for patients, especially for those with a low-performance status (59%), the choice of twice-daily therapy was mainly due to a shorter treatment duration for the patient (51%). The “concurrent” once-daily hypofractionated schedule for LS-SCLC raises concern due to the clinical feasibility [20] but many studies have verified its efficacy and safety [21,22,23]. In a retrospective study [23], hypoTRT and conventional once-daily TRT were matched by propensity score, and the results showed that hypoTRT had a comparable survival and toxicity. Socha et al. [24] declared that hypoTRT had a better survival outcome than once-daily TRT. Gronberg et al. [12] performed a randomized phase Ⅱ clinical trial including 157 LS-SCLC patients and concluded that there was no difference in the OS, PFS, and toxicity between concurrent hypoTRT (42 Gy/15 F) and the twice-daily group. Similarly, Li et al. [25] found that hypo- and hyper-IMRT resulted in comparable local control in the chest irradiation of extensive-stage SCLC. A recent phase II randomized study compared concurrent hypofractionated radiotherapy (65 Gy in 26 fractions) with concurrent twice-daily therapy in LS-SCLC, the hypofractionated radiotherapy obtained improved PFS and similar toxicities [13]. In another overlap-weighted analysis, there was no significant difference in the overall survival, local control, and toxicity for LS-SCLC patients in the hypo group versus the twice-daily group [14]. However, the efficacy of “sequential” hypoTRT in the treatment of LS-SCLC remains unclear, and we believed that patients might experience relatively better tolerance. In a retrospective study, Ohara et al. found that the efficacy of sequential chemoradiotherapy was comparable to that of concurrent chemoradiotherapy, and sequential chemoradiotherapy could be used as a treatment option for patients who were ineligible for concurrent therapy [26].

In our study, a trend was observed in terms of a longer DMFS and OS in the sequential hypo group, compared to the concurrent twice-daily group before and after PSM, although the difference was not significant. This outcome of sequential hypo was similar to previous studies [13,14,27]. In addition, the distant failure rate in our study appeared to be lower in the sequential hypo group, although there was no significant difference after PSM matching. The reason might be that the hypofractionated group received full cycles of chemotherapy ahead of time to reduce the risk of distant metastasis. The incidences of grade 1–2 esophagitis were lower in the sequential hypo group, which was due to a reduction in early response tissue damage by the hypofractionated regimen. Therefore, sequential hypo may be considered as an alternative to a concurrent twice-daily regimen in the treatment of SCLC.

In multivariable Cox regression analyses, being aged ≥65 years and having a total radiotherapy time of ≥24 days were significantly associated with a reduction in OS. The shorted radiotherapy time by the hypofractionated regimen was associated with a relatively short start of treatment to the end of radiotherapy (SER). A systematic overview [28] concluded that shortening the SER was associated with a prolonged OS. Many studies indicated that the timing of thoracic radiotherapy is a very important prognostic factor for patients with LS-SCLC [29,30,31,32,33,34,35]. In our study, an SER of >110 days was significantly associated with a reduction in OS by univariate analysis. A short SER for LS-SCLC was shown to be a good prognostic factor in some studies [29,33,34]. Being older than 65 years was also shown to be a poor prognostic factor for SCLC [36,37], which is consistent with our results. Furthermore, the doses in the hypo group varied in previous studies [11,12,21,22,24,25,38] and most of them used a hypofractionated regimen (<3 Gy). A three-dose regimens with 3 Gy per fraction was used in our study (54 Gy/18 F, *n* = 18; 45 Gy/15 F *n* = 6; 60 Gy/20 F, *n* = 5, once daily), and the optimal dose regimen needs to be investigated in further research.

Due to the retrospective nature of this study, there were the following limitations: Firstly, although PSM was performed to balance the potential confounders, the number of patients in the sample size is limited, which may cause bias in the conclusion. Secondly, the total radiotherapy time, number of chemotherapy cycles and dose/fractions were not consistent in the two groups. Therefore, a well-designed randomized control trial is needed to determine the optimal sequential hypo regimen in SCLC patients.

## 5. Conclusions

In conclusion, after propensity matching, no difference was shown in survival and failure. A sequential hypofractionated schedule was associated with comparable survival and less toxicity and may be considered as an alternative to a concurrent twice-daily regimen. Prospective trials are needed to confirm this result.

## Figures and Tables

**Figure 1 cancers-14-03920-f001:**
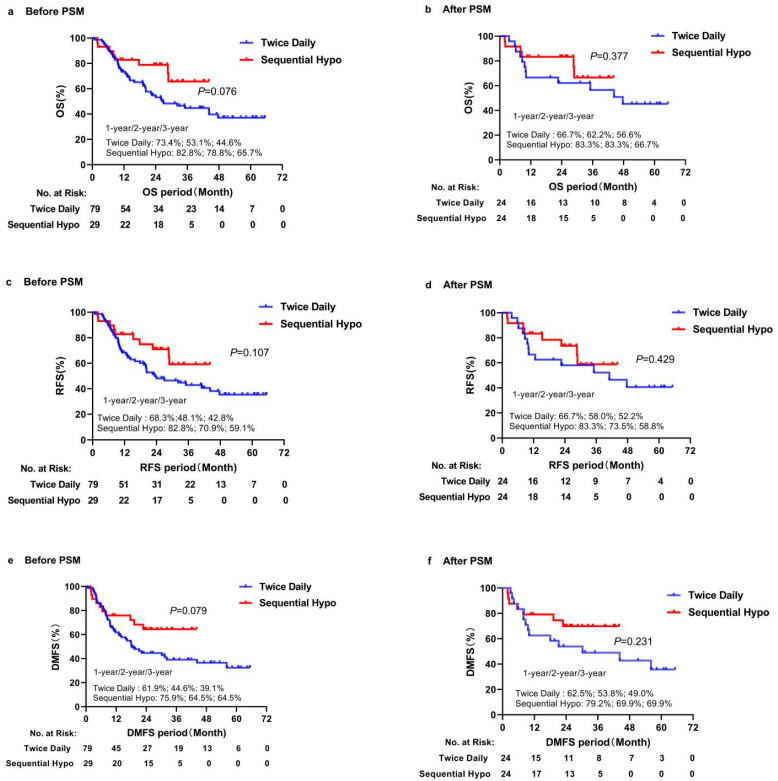
Kaplan–Meier analysis estimates the median survival time and the 1-, 2-, and 3-year OS, RFS and DMFS in different doses/fractions (sequential hypofractionated versus twice-daily): (**a**) OS before PSM; (**b**) OS after PSM; (**c**) RFS before PSM; (**d**) RFS after PSM; (**e**) DMFS before PSM; (**f**) DMFS after PSM.

**Table 1 cancers-14-03920-t001:** Baseline characteristics for patients before and after PSM [M (QL, QU)/*n* (%)].

Variables	Before PSM (*n* = 108)				After PSM (*n* = 48)			
	Total	Twice-Daily (*n* = 79)	Sequential Hypo (*n* = 29)	*p*	Total	Twice-Daily (*n* = 24)	Sequential Hypo (*n* = 24)	*p*
Age (years)	58.0 (49.0, 64.0)	57.0 (49.0, 63.0)	61.0 (52.0, 65.0)	0.171	58.5 (48.3, 64.0)	56.5 (48.3, 65.5)	59.5 (48.8, 64.0)	0.718
SER (days)	105.5 (75.3, 147.8)	96.0 (70.0, 125.0)	140.0 (106.0, 177.5)	**0.002**	128.5 (94.3, 173.8)	109.0 (77.8, 138.0)	154.5 (117.0, 195.3)	**0.006**
Total chemotherapy cycles	4.0 (3.0, 5.0)	4.0 (3.0, 5.0)	4.0 (3.5, 5.5)	0.433	4.0 (3.0, 5.8)	4.0 (3.0, 5.0)	4.0 (4.0, 6.0)	0.118
Chemotherapy cycles before radiotherapy	3.0 (2.0, 4.0)	3.0 (2.0, 4.0)	4.0 (3.0, 5.5)	**0.003**	4.0 (3.0, 5.0)	3.0 (2.0, 4.0)	4.0 (4.0, 6.0)	**0.004**
Total radiotherapy time (days)	21.0 (20.0, 25.0)	21.0 (20.0, 23.0)	24.0 (22.5, 27.5)	**<0.001**	21.0 (20.0, 25.8)	21.0 (20.0, 21.0)	23.5 (21.3, 27.0)	**0.002**
Sex				0.153				0.131
Male	78 (72.2)	60 (75.9)	18 (62.1)		31 (64.6)	18 (75.0)	13 (54.2)	
Female	30 (27.8)	19 (24.1)	11 (37.9)		17 (35.4)	6 (25.0)	11 (45.8)	
ECOG PS				0.147				1.000
0–1	91 (84.3)	69 (87.3)	22 (75.9)		42 (87.5)	21 (87.5)	21 (85.5)	
2–3	17 (15.7)	10 (12.7)	7 (24.1)		6 (12.5)	3 (12.5)	3)12.5)	
Smoking status				0.121				0.500
Never smoker	21 (19.4)	14 (17.7)	7 (24.1)		9 (18.8)	5 (20.8)	4 (16.7)	
Former smoker	36 (33.3)	23 (29.1)	13 (44.8)		20 (41.7)	8 (33.3)	12 (50.0)	
Current smoker	51 (47.2)	42 (53.2)	9 (31.0)		19 (39.6)	11 (45.8)	8 (33.3)	
PCI				**0.007**				0.755
No	28 (25.9)	15 (19.0)	13 (44.8)		15 (31.3)	7 (29.2)	8 (33.3)	
Yes	80 (74.1)	64 (81.0)	16 (55.2)		33 (68.8)	17 (70.8)	16 (66.7)	
Introduction chemotherapy				0.061				1.000
No	7 (6.5)	3 (3.8)	4 (13.8)		4 (8.3)	2 (8.3)	2 (8.3)	
Yes	101 (93.5)	76 (96.2)	25 (86.2)		44 (91.7)	22 (91.7)	22 (91.7)	
SER (day)				**0.009**				**0.027**
<90	36 (33.3)	32 (40.5)	4 (13.8)		9 (18.8)	8 (33.3)	1 (4.2)	
≥90	72 (66.7)	47 (59.5)	25 (86.2)		39 (81.3)	16 (66.7)	23 (95.8)	
Total radiotherapy time (days)				**0.001**				**0.035**
<24	74 (68.5)	61 (77.2)	13 (44.8)		31 (64.6)	19 (79.2)	12 (50)	
≥24	34 (31.5)	18 (22.8)	16 (55.2)		17 (35.4)	5 (20.8)	12 (50)	
Total chemotherapy cycles (times)				0.319				0.051
<4	34 (31.5)	27 (34.2)	7 (24.1)		13 (27.1)	10 (41.7)	3 (12.5)	
≥4	74 (68.5)	52 (65.8)	22 (75.9)		35 (72.9)	14 (58.3)	21 (87.5)	
Chemo cycles before radiotherapy (times)				0.519				
<2	15 (13.9)	12 (15.2)	3 (10.3)		3 (6.3)	3 (12.5)	0 (0)	0.233
≥2	93 (86.1)	67 (84.8)	26 (89.7)		45 (93.7)	21 (87.5)	24 (100)	
T stage				**0.035**				0.149
1–2	86 (79.6)	59 (74.7)	27 (93.2)		46 (95.8)	24 (100.0)	22 (91.7)	
3–4	22 (20.4)	20 (25.3)	2 (6.8)		2 (4.2)	0 (0)	2 (8.4)	
N stage				0.304				0.712
0–1	22 (20.4)	18 (22.8)	4 (13.8)		9 (18.8)	5 (20.8)	4 (16.7)	
2–3	86 (79.6)	61 (77.2)	25 (86.2)		39 (81.2)	19 (79.2)	20 (83.3)	
AJCC stage				0.389				0.466
I	2 (1.9)	1 (1.3)	1 (3.4)		1 (2.1)	0 (0)	1 (4.2)	
II	19 (17.6)	16 (20.2)	3 (10.3)		8 (16.7)	5 (20.8)	3 (12.5)	
III	87 (80.5)	62 (78.5)	25 (86.2)		39 (81.2)	19 (79.2)	20 (83.3)	
Comorbidity				0.091				0.477
No	83 (76.8)	64 (81.0)	19 (65.5)		38 (79.2)	20 (83.3)	18 (75.0)	
Yes	25 (23.2)	15 (19.0)	10 (34.5)		10 (20.8)	4 (16.7)	6 (25.0)	
Dose and Fractionation (Gy/F)				-				-
54/18(QD)	18 (16.7)	-	18 (62.1)		14 (29.2)	-	14 (58.3)	
45/15(QD)	6 (5.5)	-	6 (20.6)		5 (10.4)	-	5 (20.9)	
60/20(QD)	5 (4.6)	-	5 (17.2)		5 (10.4)	-	5 (20.8)	
45/30(BID)	79 (73.1)	79 (100.0)	-		24 (50.0)	24 (100.0)	-	

Abbreviations: ECOG PS: Eastern Cooperative Oncology Group Performance Status; PCI, prophylactic cranial irradiation; and SER, from the date of the start of treatment to the end of radiotherapy; *p* values less than 0.05 are highlighted in bold.

**Table 2 cancers-14-03920-t002:** Univariable Cox analysis of OS, RFS, and DMFS.

Variables	OS		RFS		DMFS	
	HR (95%CI)	*p*	HR (95%CI)	*p*	HR (95%CI)	*p*
Age (years)	1.032 (0.987–1.079)	0.163	1.038 (0.996–1.081)	0.079	1.022 (0.981–1.064)	0.303
SER (days)	1.001 (0.997–1.006)	0.564	1.001 (0.996–1.006)	0.670	0.999 (0.994–1.004)	0.754
Total chemotherapy cycles (times)	1.084 (0.805–1.460)	0.595	1.148 (0.863–1.527)	0.344	1.029 (0.782–1.354)	0.839
Chemotherapy cycles before radiotherapy (times)	1.237 (0.893–1.713)	0.200	1.218 (0.906–1.638)	0.192	1.051 (0.789–1.401)	0.735
Total radiotherapy time (days)	1.034 (0.978–1.092)	0.239	1.027 (0.973–1.084)	0.331	1.030 (0.971–1.093)	0.331
Group						
Twice-daily	1.000		1.000		1.000	
Sequential hypo	0.632 (0.228–1.748)	0.377	0.697 (0.283–1.715)	0.432	0.569 (0.223–1.449)	0.237
Age (years)						
<65	1.000		1.000		1.000	
≥65	4.483 (1.713–11.729)	**0.002**	3.528 (1.408–8.842)	**0.007**	3.383 (1.348–8.493)	**0.009**
Sex						
Male	1.000		1.000		1.000	
Female	0.783 (0.275–2.233)	0.648	0.766 (0.294–1.994)	0.585	0.567 (0.206–1.559)	0.272
ECOG PS						
0–1	1.000		1.000		1.000	
2–3	1.192 (0.454–3.129)	0.722	1.403 (0.185–10.636)	0.743	0.879 (0.117–6.602)	0.900
Smoking status						
Never smoker	1.000		1.000		1.000	
Former smoker	1.055 (0.262–4.244)	0.940	1.333 (0.352–5.050)	0.672	0.929 (0.269–3.211)	0.908
Current smoker	1.853 (0.498–6.898)	0.358	2.062 (0.565–7.525)	0.273	1.555 (0.486–4.974)	0.457
PCI						
No	1.000		1.000		1.000	
Yes	1.151 (0.408–3.247)	0.790	1.078 (0.417–2.788)	0.877	0.839 (0.366–2.092)	0.706
Introduction chemotherapy						
No	1.000		1.000		1.000	
Yes	23.292 (0.018–30,250.428)	0.389	2.047 (0.274–15.290)	0.485	23.925 (0.050–11,441.100)	0.313
SER (day)						
<110	1.000		1.000		1.000	
≥110	3.124 (1.003–9.733)	**0.049**	2.586 (0.937–7.141)	0.067	1.589 (0.629–4.016)	0.327
Total radiotherapy time (days)						
<24	1.000		1.000		1.000	
≥24	2.848 (1.105–7.344)	**0.030**	2.478 (1.039–5.908)	**0.041**	1.990 (0.808–4.901)	0.135
Total chemotherapy cycles (times)						
<6	1.000		1.000		1.000	
≥6	0.598 (0.173–2.071)	0.417	0.915 (0.334–2.509)	0.863	0.720 (0.241–2.149)	0.556
Chemotherapy cycles before radiotherapy (times)						
<4	1.000		1.000		1.000	
≥4	2.033 (0.709–5.831)	0.187	1.736 (0.669–4.507)	0.257	1.135 (0.461–2.793)	0.782
T stage						
1–2	1.000		1.000		1.000	
3–4	0.046 (0–4612.460)	0.601	1.465 (0.193–11.110)	0.712	0.046 (0–898.256)	0.542
N stage						
0–1	1.000		1.000		1.000	
2–3	2.024 (0.465–8.814)	0.348	2.576 (0.599–11.077)	0.204	2.488 (0.579–10.701)	0.221
AJCC stage						
I–II	1.000		1.000		1.000	
III	2.024 (0.465–8.814)	0.348	2.576 (0.599–11.077)	0.204	2.488 (0.579–10.701)	0.221
Comorbidity						
No	1.000		1.000		1.000	
Yes	1.274 (0.453–3.579)	0.646	0.988 (0.362–2.701)	0.982	1.234 (0.477–3.193)	0.665

Hazard ratios and 95% confidence intervals were calculated by a stratified Cox proportional hazards model. Abbreviations: OS, overall survival; RFS, recurrence-free survival; DMFS, distant metastasis-free survival; ECOG PS: Eastern Cooperative Oncology Group Performance Status; PCI, prophylactic cranial irradiation; and SER, from the date of the start of treatment to the end of radiotherapy; *p* values less than 0.05 are highlighted in bold.

**Table 3 cancers-14-03920-t003:** Multivariable Cox analysis of OS, RFS, and DMFS after PSM.

Variables	OS		RFS		DMFS	
	HR (95%CI)	*p*	HR (95%CI)	*p*	HR (95%CI)	*p*
Age (years)						
<65	1.000		1.000		1.000	
≥65	4.222 (1.601–11.134)	**0.004**	3.371 (1.337–8.498)	**0.010**	3.383 (1.348–8.493)	**0.009**
Total radiotherapy time (days)						
<24	1.000		1.000			
≥24	2.671 (1.018–7.009)	**0.046**	2.370 (0.984–5.707)	**0.054**		

Hazard ratios and 95% confidence intervals were calculated by a stratified Cox proportional hazards model. Abbreviations: OS, overall survival; and RFS, recurrence-free survival; DMFS, distant metastasis-free survival; *p* values less than 0.05 are highlighted in bold.

**Table 4 cancers-14-03920-t004:** Incidence and site of progression or recurrent disease [*n* (%)].

Site	Before PSM (*n* = 108)			After PSM (*n* = 48)		
	Twice-Daily (*n* = 79)	Sequential Hypo (*n* = 29)	*p*	Twice-Daily (*n* = 24)	Sequential Hypo (*n* = 24)	*p*
Local and/or regional						
Local only	4 (5.1)	2 (6.9)	1.000	2 (8.3)	2 (8.3)	1.000
Local and regional	2 (2.5)	0 (0)	0.952	0 (0)	0 (0)	1.000
Local and distant	5 (6.3)	2 (6.9)	1.000	1 (4.2)	1 (4.2)	1.000
Local, regional and distant	1 (1.3)	0 (0)	1.000	0 (0)	0 (0)	1.000
Regional only	3 (3.8)	0 (0)	0.686	0 (0)	0 (0)	1.000
Regional and distant	2 (2.5)	1 (3.5)	1.000	0 (0)	1 (4.2)	1.000
Distant						
Bone only	4 (5.1)	0 (0)	0.509	1 (4.2)	1 (4.2)	1.000
Liver only	1 (1.3)	1 (3.5)	1.000	1 (4.2)	1 (4.2)	1.000
Lung only	2 (2.5)	1 (3.5)	1.000	1 (4.2)	1 (4.2)	1.000
Brain only	13 (16.5)	1 (3.5)	0.144	3 (12.5)	0 (0)	0.233
Multiple location ^a^	4 (5.1)	1 (3.5)	1.000	3 (12.5)	1 (4.2)	0.602
Other location ^b^	5 (6.3)	2 (6.9)	1.000	1 (4.2)	3 (12.5)	0.602
Total locoregional failure	14 (17.7)	5 (17.2)	0.954	3 (12.5)	3 (12.5)	1.000
Total distant failure	32 (40.5)	5 (17.2)	0.024	9 (37.5)	6 (25.0)	0.350

^a^ Either combinations of bone, brain, liver and lung. ^b^ Including adrenal gland(s), eye(s), spleen.

**Table 5 cancers-14-03920-t005:** Acute and late toxicities during treatment before and after PSM [*n* (%)].

Toxicities	Before PSM				*p*	After PSM				*p*
	Twice-Daily (*n* = 79)		Sequential Hypo (*n* = 29)			Twice-Daily (*n* = 24)		Sequential Hypo (*n* = 24)		
	Grade 1–2	Grade 3–4	Grade 1–2	Grade 3–4		Grade 1–2	Grade 3–4	Grade 1–2	Grade 3–4	
Acute										
esophagitis	71 (89.9)	0 (0)	23 (79.3)	0 (0)	0.261	22 (91.7)	0 (0)	19 (79.2)	0 (0)	**0.001**
Hematotoxicity	30 (38.0)	20 (25.3)	11 (37.9)	1 (3.4)	**0.022**	11 (45.8)	6 (25.0)	10 (41.7)	1 (4.2)	0.062
Gastrointestinal	5 (6.3)	0 (0)	0 (0)	0 (0)	-	1 (4.2)	0 (0)	0 (0)	0 (0)	-
Late										
pneumonitis	3 (3.8)	0 (0)	1 (3.4)	0 (0)	1.000	1 (4.2)	0 (0)	1 (4.2)	0 (0)	1.000

Acute adverse events: ≤3 months after completion of study treatment); Late adverse events: >3 months after study treatment; *p* values less than 0.05 are highlighted in bold.

## Data Availability

All original data will be made available upon request.

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
