# Peer review of "Sequential Hypofractionated versus Concurrent Twice-Daily Radiotherapy for Limited-Stage Small-Cell Lung Cancer: A Propensity Score-Matched Analysis"

_cancers, 2022, doi:10.3390/cancers14163920_

Round 1

Reviewer 1 Report

In this manuscript, the authors analyzed the differences between sequential hypofractionated and concurrent twice daily radiotherapy on LS-SCLC patients. They followed up with 108 clinical patients who accepted the two treatments and found the patients in the sequential-Hypo schedule group have better survival and suffer less toxicity. The statistical methods performed in the data analysis are appropriate and the results are reliable. However, there are still some questions that should be clarified and explained before the publication:

1.     The supplemental figures should be mentioned in the text and the figure legends of these figures should also be replenished.

2.     The legend of table 2 has a mistake “Ha*zard ratios”, which makes the reader confused, please check and correct it.

3.     The irradiation parameters and information of the machine used should be supplied, such as the type of beam, the voltage, and dose rate of the radiation.

4.     In the statistical analysis part, the author mentioned the ROC curve, but there is no relative data shown in the text.

5.     The English needs to be edited moderately.

Author Response

Dear Reviewer:

Thank you for your precious comments and advices.We have studied comments carefully and have made corrections which we hope meet with approval. Please see the attachment.

Reviewer 2 Report

The authors compared the survival, efficiency, and toxicity of two protocols of radiotherapy in 108 SCLC. Manuscript is well written and the debate that led to the need was comprehensively described. However,

1-The final message in the abstract is confusing. Propensity matching is the most important to adjust the confounders and the output after matching should eliminate the results before matching. Despite showing primary significant results, after propensity matching, no difference was shown in survival and failure. That should be highlighted in the the conclusion to avoid misleading.

2-What other concomitant or prior treatments received before radiotherapy.

3-Some tables like the regression ones could be converted to figures for easier interpretation and capturing the results.

4-Abbreviation should be mentioned first place, e.g in lines 58, 134, and 151.

Author Response

Dear Reviewer:

Thank you for your precious comments and advices. Those comments are all valuable and very helpful for revising and improving our paper, We have studied comments carefully and have made corrections which we hope meet with approval. Please see the attachment.
